# Porin from Marine Bacterium *Marinomonas primoryensis* KMM 3633^T^: Isolation, Physico-Chemical Properties, and Functional Activity

**DOI:** 10.3390/molecules25143131

**Published:** 2020-07-08

**Authors:** Olga D. Novikova, Valentina A. Khomenko, Natalia Yu. Kim, Galina N. Likhatskaya, Lyudmila A. Romanenko, Ekaterina I. Aksenova, Marina S. Kunda, Natalia N. Ryzhova, Olga Yu. Portnyagina, Tamara F. Solov’eva, Olga L. Voronina

**Affiliations:** 1G.B. Elyakov Pacific Institute of Bioorganic Chemistry, Far Eastern Branch, Russian Academy of Sciences, pr. 100 let Vladivostoku, 159, Vladivostok 690022, Russia; homenko.40@mail.ru (V.A.K.); natalya_kim@mail.ru (N.Y.K.); galin@piboc.dvo.ru (G.N.L.); lro@piboc.dvo.ru (L.A.R.); odd64@mail.ru (O.Y.P.); soltaf@mail.ru (T.F.S.); 2N. F. Gamaleya National Research Center for Epidemiology and Microbiology, Ministry of Health of Russia, Gamaleya Str., 18, Moscow 123098, Russia; aksenova16@gmail.com (E.I.A.); markunda99@gmail.com (M.S.K.); rynatalia@yandex.ru (N.N.R.)

**Keywords:** marine bacteria, whole genome sequence, porin, amino acids composition, bilayer lipid membrane, pore-forming activity, spatial structure

## Abstract

*Marinomonas primoryensis* KMM 3633^T^, extreme living marine bacterium was isolated from a sample of coastal sea ice in the Amursky Bay near Vladivostok, Russia. The goal of our investigation is to study outer membrane channels determining cell permeability. Porin from *M. primoryensis* KMM 3633^T^ (MpOmp) has been isolated and characterized. Amino acid analysis and whole genome sequencing were the sources of amino acid data of porin, identified as Porin_4 according to the conservative domain searching. The amino acid composition of MpOmp distinguished by high content of acidic amino acids and low content of sulfur-containing amino acids, but there are no tryptophan residues in its molecule. The native MpOmp existed as a trimer. The reconstitution of MpOmp into black lipid membranes demonstrated its ability to form ion channels whose conductivity depends on the electrolyte concentration. The spatial structure of MpOmp had features typical for the classical gram-negative porins. However, the oligomeric structure of isolated MpOmp was distinguished by very low stability: heat-modified monomer was already observed at 30 °C. The data obtained suggest the stabilizing role of lipids in the natural membrane of marine bacteria in the formation of the oligomeric structure of porin.

## 1. Introduction

Gram-negative bacteria attract attention because of its extraordinary adaptability to natural environment as well as to anthropogenic, including nosocomial conditions. Marine bacteria studied since 1960s were found to be the most psychro-, piezo-, and halo-tolerant [1]. Moreover marine bacteria inhabit in oligotrophic environment with deficiency or lack of sunlight. Investigation of these bacteria revealed pressure regulation of the proteins expression [2], multiplication of genes important for metabolic pathways, and cell motility and other molecular mechanisms of the adaptation to extreme conditions [3].

Detection of the many carbon and energy utilization pathways in marine bacteria prompted the researchers to study on the properties of these bacteria as a tool for degradation of environmental pollution. Every year more attention is paid to this problem. Oil and petroleum products are the most common pollutants that disrupt and inhibit all life processes. These products accumulate as difficult-oxidized forms of substances that change the direction of metabolism and the natural ratio of the number of microorganisms in ecosystems. This leads to a slowdown in the process of microbial purification of ocean waters.

One of the most promising areas for investigating this problem is the Arctic Ocean, the smallest of the oceans which is almost completely surrounded by earth. For this reason, it has the most extensive shelf areas in comparison with any ocean basin and a significant proportion of terrigenous organic carbon flowing through rivers into the Arctic Ocean [4]. In order to determine the diversity of bacteria involved in bacterial consortia that are capable of degrading hydrocarbons, some studies were conducted in the Chukchi Plateau region [5]. Authors showed that a number of species are involved in this process. So, the potential degraders including *Cycloclasticus, Pseudomonas*, *Halomonas*, *Pseudoalteromonas*, *Marinomonas*, *Bacillus*, *Colwellia*, *Acinetobacter*, *Alcanivorax*, *Salinisphaera,* and *Shewanella*, with *Dietzia* as the most abundant, occurred in all sediment samples.

Marine sediment and water samples are the source for peptide-based drug discovery. Non-ribosomal peptides from Proteobacteria have recently attracted much attention as a source of new drugs [6]. Currently, there is considerable research interest in determining the chemical features of major components of cell envelope of marine bacteria, first of all, lipopolysaccharide (LPS) [7]. Lipid A, which is a structural component of the LPS of some marine bacteria, is considered an antagonist to endotoxins of gram-negative bacteria [8].

This is due to the fact that, in comparison with terrestrial bacteria, biologically active compounds of which are successfully used in the pharmaceutical industry, the biological potential of marine bacteria in this regard has not been practically realized [9].

However, the penetration process of the marine bacteria cells is the most intriguing. Gram-negative bacteria are covered by two distinct biological membranes. A well-established response of poikilothermic organisms to low temperature and high pressure exposure is a change in membrane fluidity [10]. There are two major classes of membrane transport proteins: transporters and channels, which are influenced by this change. Outer membrane (OM) channels include porin superfamily. These outer membrane channels share a beta-barrel structure that differ in strand and shear number. Porin superfamily comprises classical (gram-negative) porins, maltoporin-like channels, and ligand-gated protein channels cooperating with a TonB-associated inner membrane complex [11].

Classical (gram-negative) porins are known as OmpF, OmpC, and PhoE in *Escherichia coli.* To date, significant progress has been achieved in the study of the structure and function of porins of the terrestrial gram-negative bacteria, mainly members of the family Enterobacteriaceae [12,13]. However, classical porins are of the greatest interest in the context of the mechanism that mediates uptake of small molecules, including antibiotics [13].

OM porins of marine bacteria play an important role in the adaptation to the extreme environment. Few studies shed light on the role and functioning of porins of aquatic bacteria, including marine bacteria. The consequence of structural changes in the porins may be reduced permeability of the bacterial membrane for solutes. For example, in aquatic inhabitant of *Pseudomonas aeruginosa*, the permeability of the outer membrane is lower in comparison of *E. coli* [14]. Porins of some marine bacteria are sensitive to osmoregulation such as *Vibrio parahaemolyticus* and *V. alginolyticus* [15,16]. In this regard, the porins of marine bacteria are an interesting object of study of the adaptability of microorganisms.

The present study is a continuation of a series of our investigations that are dedicated to the isolation and characterization of functional activities of porins from the OM of marine microorganisms [17,18]. In this study, namely, we have isolated and characterized functional activity and spatial structure of porin from OM of *Marinomonas primoryensis* KMM 3633^T^. Our interest in this species is due to the fact that the genus *Marinomonas* includes gram-negative bacterial strains common in various marine environments. [19]. In addition, the most recognized species of the genus were isolated from sea water samples collected from different geographical locations [20]. Some of them were isolated from cold media, such as the sub Antarctic regions, for example *Marinomonas polaris* [21] and *M. ushuaiensis* [22].

During the study of microbial communities associated with marine environments in the Sea of Japan, two bacterial strains were isolated from a coastal sea-ice sample, obtained from a sea-ice column at a depth of 0.8 m in Amursky Bay near Vladivostok, Russia, in March 2001 [23]. The isolates were aerobic, gram-negative, heterotrophic microorganisms with a respiratory metabolism, and had phenotypic characteristics similar to those of the genus *Marinomonas*. They were classified as *Marinomonas primoryensis* sp. nov. [23]. *M. primoryensis* KMM 3633^T^ (= CIP 108051 = IAM 15010) is the type strain of the species. This strain was used in the present study for further investigation.

## 2. Results

### 2.1. Isolation Procedure of OM Porin from M. primoryensis KMM 3633^T^

#### 2.1.1. Cultivation Condition of Microbial Cells

Bacterial cells of *M. primoryensis* KMM 3633^T^ were grown at a temperature of 0, 6–8, and 24 °C using the growth medium described in [23]. A comparative analysis of the protein profiles of cell lysates obtained by ultrasonic disintegration was performed using SDS-PAGE (data not shown). Significant differences in the polypeptide composition of cell lysates from bacteria grown at a reduced temperature (0 and 6–8 °C) was not revealed. However, cultivation at room temperature led to some decrease in the intensity of polypeptide bands in the region of high molecular weight proteins. In accordance with the results obtained, for further studies, the microbial cells of M. primoryensis KMM 3633^T^ grown at a temperature of 6–8 °C were used.

#### 2.1.2. Isolation of Cell Envelope of *M. primoryensis* KMM 3633^T^

To remove surface proteins, microbial cells of *M. primoryensis* KMM 3633^T^ were pre-treated with 5% solution of NaCl. Then, to obtain the cell envelope from the harvested microbial cells ultrasonic disintegration was used. The protein profile of the resulted total crude membrane fraction consisted of several major components: high molecular weight (MW) proteins in the form of a diffuse zone with apparent MW in the range of 95–130 kDa and a number of polypeptides with a MW in the range of 34–72 kDa (Figure 1A, lane 2). Judging by the intensity of the polypeptide bands their content in this area of molecular masses varies greatly. To detect heat-modifiable proteins in the envelope fraction, boiling in denaturing conditions was used (Figure 1A, lane 3). Several polypeptide zones in the region of 34–55 kDa became more clearly visible in MW ladder, while the protein bands in the region of high MW proteins disappeared. In addition, the intensity of the polypeptide band with MW between 43 and 35 kDa increased significantly. Obviously, the oligomeric polypeptide region of cell lysate of *M. primoryensis* KMM 3633^T^ includes several heat-modifiable proteins which are supposed to be porins. Such behavior in the denaturing conditions of SDS-PAGE is typical for OM porins of gram-negative terrestrial bacteria, in which the oligomers dissociate into monomers after heating above the critical temperature of an irreversible conformational transition. To obtain isolated proteins from the crude cell envelope fraction of *M. primoryensis* KMM 3633^T^ two methods were used: Sarcosyl treatment and octyl-POE extraction.

#### 2.1.3. Sarcosyl Treatment

Extraction of the crude cell membranes of *M. primoryensis* KMM 3633^T^ with 1% solution of Sarcosyl [24] resulted in an insoluble fraction of OM proteins. The precipitate obtained after washing with HEPES was a pure sample of the protein oligomer, whose electrophoretic mobility corresponded to MW about 95 kDa (Figure 1B, lane 1). The electrophoretic mobility of the monomer of the target protein under denaturing conditions corresponded to a polypeptide with a molecular weight of about 35 kDa (MpOmp) (Figure 1B, lane 2).

#### 2.1.4. Octyl-POE Extraction

To obtain the fraction with the highest content of heat-modifiable OM proteins, a stepwise extraction of the cell walls of *M. primoryensis* KMM 3633^T^ with a POE solution was carried out, as is known, selectively extracting OM proteins of gram-negative bacteria [25]. It was found that polypeptides changing their molecular weight when heated under denaturing conditions are extracted at different concentrations of POE in the range from 0.5 to 3.0% by weight [25]. In the case of *M. primoryensis* KMM 3633^T^, the greatest amount of the heat-modifiable proteins was obtained using 0.5% detergent solution, so the extraction was repeated three times. According to the SDS-PAGE data an electrophoretically homogeneous protein was isolated in the oligomeric form as a result of purification of the obtained sum fraction by gel permeation chromatography on Sephacryl S-300 (Figure 1B, lane 4). After boiling in SDS solution the protein band observed migrated to the same position as MpOmp monomer isolated with Sarcosyl extraction (Figure 1B, lane 3).

It should be noted that in this case, the unheated protein migrated to a position corresponding to MW of about 72 kDa. This apparent MW value was lower than that of the MpOmp oligomer isolated by Sarcosyl extraction. The cause of this phenomenon could be the presence of residual amount of POE in the protein sample. A similar shift of the protein MW was observed for VhOmp of *Vibrio harveyi* in [26].

Thus, according to the SDS-PAGE data, MW the protein isolated with Sarcosyl extraction was identical to that of the protein obtained by the Garavito method [25]. Taking into account the data obtained, in subsequent experiments a sample obtained as a result of the removal of cytoplasmic proteins with Sarcosyl was used. We chose this method of the target protein isolation because of the simplicity and at the same time the high selectivity of the isolation procedure.

### 2.2. Pore-Forming Activity of OM Porin from M. primoryensis KMM 3633^T^

The observed pore-forming activity of the total fraction of OM proteins of *M. primoryensis* KMM 3633^T^ was instable; nevertheless, this confirmed the presence of porin among the OM proteins of the bacteria (data not shown). When purified MpOmp was added in small quantities (10–100 ng/mL) to the aqueous solution bathing an artificial lipid bilayer, the membrane conductance increased by several orders of magnitude (Figure 2A,B). The current-voltage characteristic of the porin channel is linear in the range up to 180 mV (Figure 2C). The pore-forming activity of MpOmp was found to depend on the salt concentration in the aqueous phase. We failed to record channel formation at NaCl concentration below 0.2 M. The linear dependence of the current through the porin channel is observed when the salt concentration varied in the range 0.5–2.0 M (Figure 2D).

BLM formed from a 1% solution of 1-monooleoylglycerol in *n*-heptane. Aqueous phase: 20 mM Tris-HCl, pH 7.4 (buffer A), 0.5–2.0 M NaCl, protein concentration 10–100 ng/mL. Membrane potential is 50 mV.

### 2.3. Amino Acid Composition and Amino Acid Sequence of OM Porin from M. primoryensis *KMM 3633^T^*

*M. primoryensis* KMM 3633^T^ whole genome was sequenced, assembled, and annotated as described in Methods. The porin of *M. primoryensis* KMM 3633^T^ was classified as Porin_4 according to the conservative domain searching and analyzed in detail. The characteristics are represented in Table 1. By protein BLAST we revealed proteins with high sequences similarity and proteins with similar domain architecture. Porins of *M. primoryensis* strain AceL and *Marinomonas* sp. strain IMCC 4694 had the highest similarity with porin of strain KMM 3633^T^-83.59% and 77.43%, respectively. Porins of *Polaromonas sp*. JS666, *Magnetospirillum magneticum* AMB-1 and *Basfia succiniciproducens* MBEL55E had the most closely related domain architecture with analyzed porin. Porins of *Photobacterium damselae* [27] and *Escherichia coli* [28] were chosen as well-known porins: the first one is from marine Gammaproteobacteria, the fish pathogen, and the second one is from terrestrial Gammaproteobacteria, the classical object of biological and molecular research. Amino acid composition of the isolated proteins [27,28] was supplemented by translation of sequenced genes.

Most of the bacteria represented in Table 1 are aquatic, isolated from sea-, ground-, or bog water. *E. coli* and *B. succiniciproducens* are known as habitants of gastrointestinal tract (GIT). All eight analyzed bacteria were Proteobacteria, six of them were from class Gammaproteobacteria. According to the Pmaf classification porins of aquatic bacteria belong to one superfamily, and GIT bacteria—to another. According to TBCD (Transporter Classification DataBase) all porins belonged to the same superfamily and all but one were assigned to the same family. As follows from the data shown in Table 2, eight porins were not much different in number of amino acids, protein molecular weight, and aliphatic index, but significant differences were revealed in theoretical pI, total number of negatively and positively charged residues, and differences in the content of tryptophan residues. This amino acid was absent in the porins of the three *Marinomonas* sp. strains and *Photobacterium damselae subsp. damselae* and was in low amount in porin of *Polaromonas* sp. JS666 in comparison with porins of the GIT bacteria. All porins analyzed are enriched in polar amino acids and have a significantly higher level of acidic amino acids (15.9–28.8 mol %) compared with the basic amino acids (5.1–16.4 mol %), that is typical for classical gram-negative porins. Despite noted differences all porins had close polarity values calculated by two methods.

The polarity of MpOmp was 44.25 (from amino acid composition) and 50.7 (from DNA translation) calculated as reported previously [29]. The polarity index of the protein studied was 1.81 (from amino acid composition) and 1.95 (from DNA translation) calculated by [30]. The value is close to the average that of similar values for other proteins belonging to the Protein_4 superfamily represented in Table 1. This result and GRAVY index (Table 1) indicate that porin from *M. primoryensis* KMM 3633^T^ is relatively hydrophilic protein.

ProtParam tool allows the computation of the instability index on the base of dipeptide composition. Since the proteins with instability index value below 40 are stable proteins [31], all porins from our analysis are stable, and porins of *M. primoryensis* KMM 3633^T^ and *M. primoryensis* AceL was predicted to be the most stable (Table 1).

### 2.4. Spatial Structure of OM Porin from M. primoryensis KMM 3633^T^

#### 2.4.1. CD and Intrinsic Protein Fluorescence Spectra

One of the important physicochemical characteristics of proteins is their resistance to various denaturing agents and temperature. Therefore, we paid special attention to conformational changes in MpOmp molecule under various conditions: (1) upon dilution of the protein solution, (2) in the presence of non-ionic (POE) and ionic (SDS) detergents, and (3) upon heating. For this purpose, CD analysis and intrinsic protein fluorescence were used. CD spectra were recorded in the far and near UV regions. Intrinsic protein fluorescence spectra were recorded at an excitation wavelength of 280 nm.

It was found that conformational changes were not observed either at the level of the secondary or at the level of the tertiary structure of MpOmp upon dilution of the protein solution without any detergents (Figure 3A,B).

CD spectra were recorded in the far and near UV regions. CD spectra of MpOmp in the aromatic region (240–320 nm) have a low amplitude and low resolution, which indicates a loosened tertiary structure because of the apparently weak interaction between the protein monomers (data not shown).

CD spectra of the porin studied between 180–240 nm (the region of peptide bonds) in the presence of non-ionic detergent POE are characterized by positive band at 195 nm and only one negative minimum that was centered at 220 nm. The spectra crossed the zero line at 209 nm (Figure 3C). It is typical for proteins with β-pleated sheet structure of (α + β) type [32]. In the presence of SDS a new negative band at 207 nm was observed (Figure 3E), which indicated α-helical structure formation typical of denatured proteins in SDS [33].

Using the CDPro software [34] the content of secondary structure elements of MpOmp was determined. As we can see in Table 3, the secondary structure of the protein in solution of various detergents differed. In POE solution the total β-structure of MpOmp accounted 70% while alpha α-helices make up only 5%. Such ratio of the elements of the regular secondary structure of the protein is similar to that of the porins in the native environment in the bacterial membrane [35].

However, in the presence of SDS, the polypeptide chain of the studied protein contains a significant amount of α-helical regions (3.6 times more than in a solution of non-ionic detergent) and a smaller amount of total β-structure. These data indicate that ionic detergent SDS has a denaturing effect on the secondary structure of MpOmp. Similar denaturing effect of this ionic detergent we observed earlier for *Yersinia* porins [36].

In Figure 3 the intrinsic fluorescence spectra of MpOmp in solutions of POE (Figure 3D) and SDS (Figure 3F) are presented. A significant shift of the maximum of the spectrum to the short- wavelength region in the presence of ionic and nonionic detergents indicates that the radiation of the protein occurs because of tyrosine residues. These results are consistent with the data of amino acid sequence (Table 2), according to which the tryptophan residues are absent in the protein studied.

Thus, according to data of optical spectroscopy the secondary structure of MpOmp depends on the detergent nature present in its solution.

#### 2.4.2. Temperature Stability of OM Porin from *M. primoryensis* KMM 3633^T^

The instability of MpOmp trimer was demonstrated by heating a protein sample at different temperatures. As follows from the data of SDS-PAGE (Figure 1C), protein samples of MpOmp, both isolated by POE extraction and obtained by treatment of the cell membrane with Sarcosyl, are equally temperature sensitive. Dissociation of the oligomeric form of the protein into monomers was observed already at a temperature of 30 °C. In order to analyze the changes in the spatial structure of porin upon heating and to differentiate the dissociation process from the process of protein denaturation, CD spectra in the peptide region and emission of MpOmp samples heated at different temperatures from 30 to 80 °C were recorded.

As follows from the data in Figure 3C–F, in MpOmp molecule in POE solution, changes at the level of the secondary structure of the protein are observed starting at a temperature of 40 °C. Further heating of the sample leads to more significant changes, manifested in a decrease in the ellipticity of characteristic peaks.

In the SDS solution in the spatial structure of the porin molecule, more serious rearrangements even occur at 30 °C, a change in the peak ratio at 195 and 207 nm is observed, which may indicate a change in the content of α-helices. With a further increase of temperature of the sample treatment, the observed changes were minimal.

The heating of MpOmp sample in solutions of both detergents in the studied temperature range did not lead to any major noticeable changes detected by fluorescence spectroscopy. The data obtained allow us to conclude that only an ionic detergent is capable to effect the conformational changes in the structure of the porin studied.

## 3. Discussion

### 3.1. Isolation Procedure of OM Protein from M. primoryensis *KMM 3633^T^*

To date several advances have been made in the development of isolation methods of OM pore-forming proteins. Since porins as integral proteins exist in an intrinsically anisotropic environment within the bacterial membrane the main condition for maximizing protein extraction is the selection of a suitable detergent [25]. Unlike terrestrial, marine bacteria cell envelope is not tightly associated with rigid peptidoglycan layer and therefore the OM may be easily separated [37,38]. Thus, the standard isolation and purification procedure used for OM porins of gram-negative bacteria is not exactly applicable for marine OM proteins. In the case of marine bacteria, the OM proteins may be isolated with the sequential extraction of the microbial cells with solutions of high ionic strength, even without previous cell destruction [39]. In light of the above special attention has been given by us to the choice of cell envelope isolation condition and the selection of a suitable detergent for the extraction of *M. primoryensis* KMM 3633^T^ OM proteins.

To determine the influence of the cultivation temperature on the protein composition of the cell envelope of the psychrophilic bacterium *M. primoryensis* KMM 3633^T^, the bacterial cells were grown at a temperature of 0, 6–8, and 24 °C using the growth medium given in [23]. When growing microorganisms at low temperatures, the most intense polypeptide band was observed in the region corresponding to the molecular masses of the porins of terrestrial bacteria. Taking these results into account, for the further study we selected the *M. primoryensis* KMM 3633^T^ cells, cultivated at 6–8 °C under aerobic conditions.

In the electrophoretic profile of the crude membrane fraction obtained by ultrasonic disintegration, proteins that changed their electrophoretic mobility after heat pre-treatment of the protein sample were detected. This allows us to assume that heat-modifiable proteins are present in the cell lysate of *M. primoryensis* KMM 3633^T^.

To isolate the OM protein fraction from the cell envelope of *M. primoryensis* KMM 3633^T^, two methods were tested, commonly used to obtain OM proteins from the terrestrial bacteria. One of them consists in the step-by-step extraction of cell membranes obtained after ultrasound treatment with a nonionic detergent POE [25]. By the author data subsequent extraction with POE allows to yield several fractions highly enriched in *E. coli* porins.

The second method is based on the removal of cytoplasmic proteins from the cell envelope fraction with nonionic detergent Sarcosyl extraction. As known, Sarcosyl is commonly used in the purification procedure of OM proteins of gram-negative terrestrial bacteria, and this method produces samples free of cytoplasmic proteins. OM proteins in this case remain in the sediment. As we have seen earlier, after Sarcosyl pre-treatment the fraction of OM proteins of *Yersinia ruckeri* practically did not contain low molecular weight cytoplasmic proteins [40]. However, in the case of OM proteins of *M. primoryensis KMM* 3633^T^ in the Sarcosyl-soluble fraction, in addition to cytoplasmic proteins, we found the presence of a certain amount of OM proteins with a molecular weight of 30–50 kDa. This result indicates that the OM proteins of *M. primoryensis* KMM 3633^T^ are partially soluble in Sarcosyl.

### 3.2. Pore-Forming Activity of Porin M. primoryensis KMM 3633^T^

It was found that the channels of MpOmp are characterized by a linear current-voltage characteristic and the dependence of conductivity on salt concentration. Purified porin reconstituted in BLM at a concentration 10–100 ng/mL induced stepwise changes in membrane conductance typical for porins. It is noteworthy that the activity of marine porins [17,18] as a rule, is lower (at 100–200 ng/mL) as compared to the activity of Enterobacteriaceae porins (at 1–10 ng/mL) [41].

Within the transmembrane potential of ±180 mV, MpOmp pores acted as ohmic channels, and their conductivity scaled linearly with voltage. Thus, the porin from *M. primoryensis* KMM 3633^T^ behaves in a voltage-dependent manner, like the other marine porins [26,42].

It was shown that the pore-forming activity of the protein studied depends significantly on the ionic strength of the electrolyte. This is characteristic of the porins of marine bacteria. It is generally accepted that non-linear graphs of the dependence of the electrical conductivity of pores on the symmetric salt concentrations on both sides of the protein-containing membrane indicate the influence of the functionality of the internal channel on the passage of charged particles [26]. In the future we plan to build a theoretical protein model of MpOmp that will allow us to conduct a detailed analysis of charge distribution inside the barrel channel.

### 3.3. Physico-Chemical Properties of Porin from M. primoryensis *KMM 3633^T^*

The amino acid composition of porin-like protein from *M. primoryensis* KMM 3633^T^ has the same characteristic features as those of the classical gram-negative porins. For example, the protein composition is distinguished by a high content of acidic amino acids and low content of sulfur-containing amino acids. Cysteine is completely absent. However, there is a slight difference in the content of individual amino acids in composition of the protein characterized: it contains 1.4 times less aromatic acids and almost 1.4 times less tyrosine residues compared to the OmpF porin *E. coli.* In addition, the composition and properties of MpOmp are quite remarkable, primarily because of the fact that, in contrast to the porins of the other Proteobacteria under consideration (except for the porins from bacteria of the genus *Marinomonas* and *Photobacterium damselae subsp. damselae* presented in Table 1), there are no tryptophan residues in its molecule.

Considering the monomer stability of the analyzed porins, we noticed that predicted instability index had the lowest value for porins of *M. primoryensis* KMM 3633^T^ and *M. primoryensis* AceL. When characterizing the monomers of these porins as the most stable, we take into account that in the bacterial cell the stability of the protein may be dependent not only on the intrinsic nature of the protein but also on the conditions of the protein milieu, as noted Gamage, et al. in their study [43]. In connection with the foregoing, we should especially dwell on the instability of MpOmp oligomeric structure that we observe.

Porin *M. primoryensis* KMM 3633^T^ was isolated in the oligomeric form, however porin trimers were extremely unstable. The dissociation of MpOmp trimer into monomers, accompanied by an irreversible conformational transition was already observed at 30 °C. In a number of cases, on the electropherogram of the fraction of OM proteins obtained under denaturing conditions of SDS-PAGE (0.1% SDS), the protein monomer of MpOmp appeared without prior boiling of the sample. In addition, MpOmp trimer dissociated into monomers also after protein precipitation with ethanol in the presence of EDTA, the method used for purification of membrane proteins [25]. So, a trimeric form of the protein was clearly observed only under mild condition of solubilization. Therefore we cannot exclude that isolation conditions of target protein chosen by us do not lead to a partial dissociation of its trimeric form during obtaining of cell envelope and purification from cytoplasmic proteins. Thus, in order to obtain and preserve OM protein from *M. primoryensis* KMM 3633^T^ as a trimer with a relatively stable structure, it is necessary to choose only a mild non-ionic detergent and strictly monitor the temperature at which the experiment is conducted. Similar instability of native subunit structure of some bacterial pore-forming proteins was described in literature [44,45]. Authors of these articles should have used cross-linking in order to reveal the native oligomers of the proteins studied.

Elucidation of the conformational stability of the porin studied under various conditions using the methods of optical spectroscopy and electrophoresis made it possible to draw the following conclusion. It was found that no conformational changes were observed either at the level of the secondary or at the level of the tertiary structure of the protein upon dilution. However, the presence of ionic and nonionic detergents influenced the spatial organization of MpOmp differently and introduced characteristic features into the conformation of the protein molecule. It has been shown that the SDS ionic detergent has a denaturing effect on the conformation of the porin trimer from *M. primoryensis* KMM 3633^T^ at the secondary protein structure, significantly increasing the content of the α-helix and decreasing the content of the total β-structure. This is consistent with the widespread literature hypothesis that SDS micelles promote the formation of α-helices in the case of β-structured proteins, containing intrinsically disordered regions (IDRs) [46]. In contrast, in the presence of a non-ionic detergent, minimal changes in protein structure were observed. Thus, the data obtained show, that non-ionic detergent POE is most suitable for solubilization of MpOmp, since in its presence the conformation of the studied protein is as close as possible to the native one.

According to modern concepts, porins are unusual membrane proteins containing a significant amount of hydrophilic amino acid residues. They are able to form β-barrels in contrast to the α-helices of almost all other membrane proteins. The association of monomers within native trimer is stabilized through the hydrophobic and hydrophilic interaction at the subunit interface and involves 35% of the barrel surface area [47].

In addition, porins are amyloidogenic proteins capable of forming amyloid-like structures under certain conditions. This may be facilitated by the availability of hydrophobic sites on the surface of the monomer, buried in the trimer, but released during its dissociation. At the same time, according to the recently published theory of the formation of multimeric proteins, the formation of their quaternary structure occurs because of the balance between the energy of intramolecular interaction and the energy of adhesion between subunits [48].

Given these facts, it can be assumed that the instability of the trimeric form of MpOmp can be caused both by the difference in the values of the indicated energies in favor of the predominant existence of a protein in the form of a monomer, and by differences in the primary structure of porins from marine and terrestrial bacteria, the trimer form of the last exhibit the high thermal stability. So, for example, mutation of residues involved in ionic interactions between the two subunits have been shown to reduce the thermal stability of the trimer significantly [49].

At present, computer models are widely used in investigation of the stabilizing forces in quaternary structure formation. Currently, there is the possibility of a detailed study of protein–protein interaction in the formation of the oligomeric structure of beta-barrel proteins [50]. Transmembrane domains of β-barrel membrane proteins have shown the presence of so called “weakly stable regions” despite an extensive network of hydrogen bonds, as well as ionic and hydrophobic interactions that give high strength to the molecule as a whole [51]. In addition, it was found that one of the ways to stabilize these areas can be by interacting with surrounding lipids [52]. On the other hand, it is known that oligomerization of the proteins, including membrane proteins, can bring various functionally important advantages to a particular protein.

Taking into account the foregoing, we can assume the following. Since the MpOmp monomer has very high stability (Table 1), and the oligomeric structure of the isolated protein, on the contrary, is extremely unstable under the heating, it is possible that the existence of the MpOmp trimer is crucial for the manifestation of the functional activity of this porin in the native membrane. This assumption is based on the fact that isolated MpOmp in monomeric form had a very low efficiency of reconstitution into an artificial bilayer. Given the fact that lipid–protein interaction can be a factor stabilizing the trimeric structure of MpOmp, the study of the pore-forming activity of this novel marine porin depending on the composition of the lipid membrane will undoubtedly be of fundamental interest.

## 4. Materials and Methods

### 4.1. Microorganisms

The bacteria of *M. primoryensis* KMM 3633^T^ were grown at a low (0 and 6–8 °C) and room temperature (24–26 °C) in liquid medium containing sea water as described in [23].

### 4.2. Isolation of Cell Envelope Fraction

Microbial cells were harvested in the late exponential growth phase (after 48 h) that was determined by the turbidity of the cell suspension. Cells were collected by centrifugation at 7000 rpm for 20 min, washed once with a solution of 5% NaCl, and suspended in buffer containing 50 mM Na_2_HPO_4_, 100 mM NaCl, sucrose (5%, *w*/*v*), 1.5 mg of DNAase, and sodium azide (0.02%, *w*/*v*).The cells were disrupted by sonication (UZDN-2T insonator, Sumy, Ukraine) at 44 kHz (10 × 1 min cycles on ice). The unbroken cells were centrifuged at 10,000× *g* for 15 min at 4 °C, and the supernatant was centrifuged at 25,000× *g* (Heraeus Biofuge stratus, Thermo Fisher Scientific, Waltham, MA, USA) for 1 h at 4 °C. The pellet was then washed with distilled water and used as cell envelope preparation (crude membrane fraction). It contained the total fraction of cytoplasmic and OM proteins.

### 4.3. Isolation and Purification of Porin from M. primoryensis *KMM 3633^T^*

#### 4.3.1. Extraction with Nonionic Detergent *n*-Octylpolyoxyethylene (POE)

Cell envelope pellet (6.4 mg) was resuspended in 5 mL 20 mM Tris-HCl, pH 7.4 (buffer A), containing 0.5% (*v*/*v*) POE incubated subsequently at 37 °C for 1 h and at 4 °C overnight, and then centrifuged at 25,000× *g* for 1 h. This procedure was repeated three times. The pellet was washed once with buffer A by centrifugation. The residual pellet was extracted consistently three times with 3 mL of buffer A, containing 0.5, 1, and 2% (*v*/*v*) POE. According to the SDS-PAGE data the abundant quantity of 35 kDa protein (MPOmp) was found in supernatant after extraction with 0.5% POE. Three 0.5% POE extracts were combined, concentrated to the protein content of 0.5 mg/mL, and subjected to dialysis against a 10-fold volume of buffer A, containing 0.2% POE and then purified by gel permeation chromatography on Sephacryl S-300 (Serva, Heidelberg, Germany).

#### 4.3.2. Extraction with N-Lauroylsarcosine (Sarcosyl)

Cell envelope pellet (6.4 mg) was resuspended in 5 mL 10 mM HEPES, pH 7.5 (buffer B), containing 1% (*w*/*v*) Sarcosyl (Sigma-Aldrich, St. Louis, MO, USA) and incubated at room temperature for 30 min with shaking. The pellet obtained by centrifugation at 25,000× *g* was washed with 10 mL of buffer B without suspending. According to SDS-PAGE data, this fraction was electrophoretically pure 35 kDa protein (MpOmp) in oligomeric form (approximately 100 kDa).

### 4.4. SDS-PAGE

Analysis of protein fractions obtained was carried out using SDS-PAGE according to the method described in [49,50]. SDS-PAGE procedure was carried out under native (solubilization temperature 25 °C) and denaturing (after heating at 99 °C) conditions. Apparent MW of proteins in different fraction was determined according to relative mobilities of standard proteins using marker proteins (Thermo Fisher Scientific, Waltham, MA, USA) with MW from 10 to 170 kDa. The proteins separated in the gel were stained with Coomassie R-250 in 3.5% perchloric acid [53]. The protein content of the samples was determined spectrophotometrically at 280 nm, assuming an optical density of 1.0 for a protein concentration of 1 mg/mL.

### 4.5. Spectroscopic Methods

All absorption, circular dichroism (CD), and fluorescence measurements were performed using the protein samples in buffer A, containing 0.25% SDS at 25 °C. UV absorption spectra were recorded on a UV-visible spectrophotometer UV-1601PC (Shimadzu, Japan) in quartz cuvettes with the path-length of 1 cm. The correction for light scattering of the protein solution was carried out as described in [54]. Specific absorption factor A 0.1%/1 cm of porin was taken equal to 1.00.

*Circular dichroism (CD) spectra* were recorded on Chirascan Plus CD spectropolarimeter (Applied Photophysics Limited, Leatherhead, UK) in quartz cuvettes with the optical path length of 0.1 and 1 cm for the far-UV or peptide (180–250 nm) and the near-UV or aromatic (250–350 nm) regions, respectively. Samples with the protein concentrations of 1.0 µM and 6.0 µM were used for the CD measurements in the peptide and aromatic regions, respectively.

In the aromatic spectral region, ellipticity [θ]M was calculated as molar ellipticity taking the molecular weights of the protein monomer and trimer as 35 and 105 kDa, respectively. In the peptide region, ellipticity was calculated as the mean residue ellipticity taking the molecular weight of amino acid residue as 110 Da, in standard units (deg cm^2^ dmol^−1^) by the equation:[θ] = [θ]_obs_·S·110/10·C·l(1)
where S is the sensitivity; C is the protein concentration, mg/mL; and l is the optical path length, dm.

The content of the secondary structure elements of the protein was calculated with CD Pro software [34].

*Intrinsic protein fluorescence spectra* of the porin were measured in quartz cuvettes with optical path length of 1 cm using a RF-5301PC spectrofluorophotometer (Shimadzu, Kyoto, Japan). Porin samples with protein concentration of 0.5 µM were prepared in buffer A, containing 0.25% SDS. Fluorescence was excited by the light with wavelength of 280, 296, and 305 nm. The excitation and emission slits were set at 5 nm. Fluorescence spectra corrected by rhodamine B (Wako Pure Chemical Industries, Osaka, Japan) were recorded by subtracking the Raman band of the buffer solution.

### 4.6. Lipid Bilayer Experiments

The experiments on reconstitution of protein samples into the BLM were performed under the symmetric conditions, so that the investigated proteins were present on both sides of the BLM. Fluctuations of the current through the BLM in the presence of the porin were measured in the clamp mode on the membrane potential. The current from the membrane was conducted by Ag/AgCl electrodes and registered at a membrane potential of 10–50 mV. The electrical parameters of the BLM were measured with bilayer clamp amplifier BC-535, pCLAMP 10 and Clampfit 10 software suites.

The BLM was formed from a 1% solution of 1-monooleoylglycerol (monoolein) (Sigma- Aldrich, St. Louis, MO, USA) in *n*-heptane with a pipette (a hole of 0.25 mm in diameter) on a Teflon cup placed in a thermostated optically transparent cuvette filled with an electrolyte. The experiments of protein reconstitution in the BLM were performed at room temperature (22 °C). The aqueous phase contained 0.2–2 M NaCl in buffer A and the studied protein in the range of concentrations of 10–100 ng/mL.

### 4.7. Amino Acid Composition and N-Terminal Sequencing

*Amino acid analysis* was performed by using a Biochrom 30 amino acid analyzer (Biochrom Ltd. CB4 OFJ, Cambridge, UK), after hydrolysis of purified protein with 6 N HCl according to the method descried in [55]. The amino acid content of the protein sample was calculated in Mol% of each amino acid.

*DNA isolation.* Preparation of genomic DNA for whole genome sequencing (WGS) was performed according to the protocols [56].

*Genome sequencing and assembly*. Two kits were used for library preparation: KAPA HyperPlus (Roche, Basel, Switzerland) and Nextera XT DNA Library Prep Kit (Illumina, San Diego, CA, USA). Libraries were checked on a Bioanalyzer (Agilent, Santa Clara, CA, US) and sequenced on an Illumina MiSeq instrument with paired-end protocol. Genome assembly was performed with CLC Genomic Workbench v.12.

*Genome annotation*. Genome was annotated by the Rapid Annotations using Subsystems Technology (RAST) server [57,58].

GenBank NCBI submission. The obtained *Marinomonas primoryensis* KMM 3633^T^ genome sequence was submitted in NCBI SRA database under the BioProject number: PRJNA622482. DNA of Porin_4 was translated, complementary annotated in NCBI by conserved domains detection and submitted in GenBank with Accession Number MK820371 (Protein Accession-QES04118).

Sequences, used for analysis. *Marinomonas primoryensis* strain AceL (WP_112140846), *Marinomonas* sp. IMCC 4694 (WP_148833815), *Polaromonas* sp. JS666 (Q123M1), *Magnetospirillum magneticum* AMB-1 (Q2WBE6), *Photobacterium damselae subsp. damselae* (KJF82573.1), *Escherichia coli* (WP_000768382.1), *Basfia succiniciproducens* MBEL55E (Q65V42).

*Bioinformatic analysis*. Alignment of protein sequences was performed by NCBI BLAST, signaling peptide was predicted by Phobius server (http://phobius.sbc.su.se/) [59]. ProtParam tool was used for the computation of various physical and chemical parameters for the proteins (https://web.expasy.org/protparam/) [60].

Polarity index was calculated according to following formula [30]:

(Asx + Glx + Lys + Arg + Ser + Thr)/(Val + Leu + Ile + Met + Pro + Phe). For the polarity calculation was used formula recommended by [29]:

(Asx + Glx + Lys + Arg) + (Ser + Thr + Tyr + His + Gly)/2

PSORTb program was used for bacterial protein subcellular localization prediction (https://www.psort.org/psortb/) [61,62].

Pfam server was used for protein classification (http://pfam.xfam.org/) [63].

TCBD (Transporter Classification DataBase) and bioinformatics resources of this base were used for porin classification [63].

## Figures and Tables

**Figure 1 molecules-25-03131-f001:**
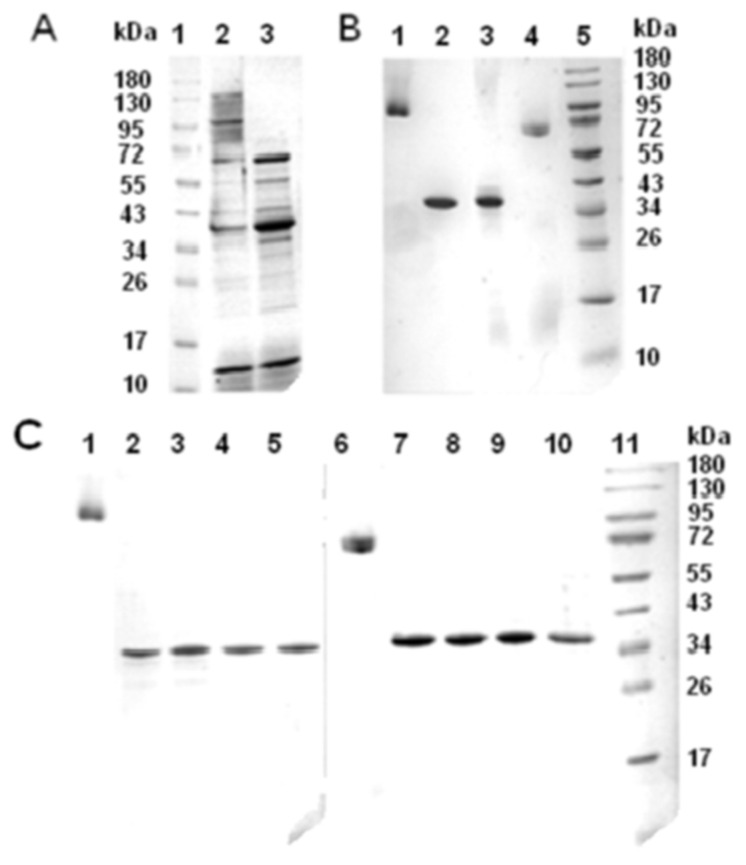
Purification and temperature denaturation of *M. primoryensis* KMM 3633^T^ porin (MpOmp): (**A**): Crude cell envelope fraction (crude membrane) of *M. primoryensis* KMM 3633^T^ (2, 3); marker proteins (1). Sample 3 is heated at 99 °C. (**B**): MpOmp isolated from the crude membrane fraction of *M. primoryensis KMM* 3633^T^ after treatment with 1% Sarcosyl (1, 2); MpOmp isolated from combined OM protein fraction obtained as a result of three time extraction with 0.5% POE of *M. primoryensis* KMM 3633^T^ cell envelope and gel permeation chromatography on Sephacryl S-300 (3, 4); marker proteins (5). Samples 2 and 3 are heated at 99 °C. (**C**): MpOmp isolated with Sarcosyl extraction (1–5); samples 2, 3, 4, and 5 were heated at 30, 40, 50, and 99 °C, respectively; MpOmp isolated with POE extraction (6–10); samples 7, 8, 9, and 10 were heated at 30, 40, 50, and 99 °C, respectively; marker proteins (11).

**Figure 2 molecules-25-03131-f002:**
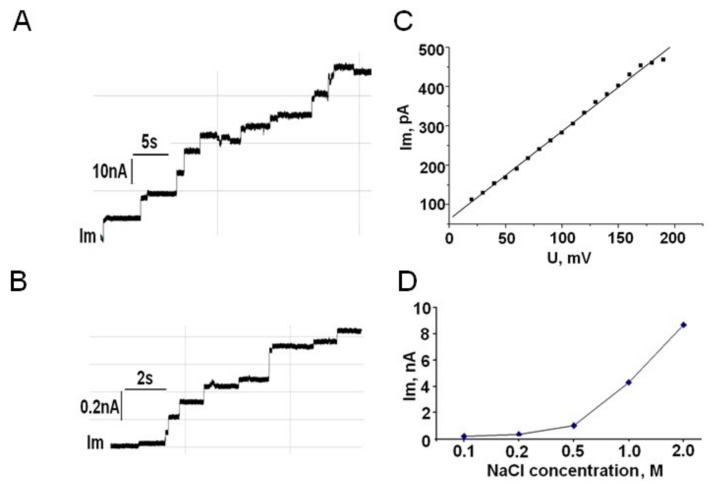
Current traces recorded after the addition of *M. primoryensis* KMM 3633^T^ porin to bilayer lipid membrane (BLM) in the presence of 1.0 M (**A**) and 0.2 M (**B**) NaCl; current-voltage characteristic of MpOmp channel (**C**); the dependence of the current through the MpOmp channel on NaCl concentration (**D**).

**Figure 3 molecules-25-03131-f003:**
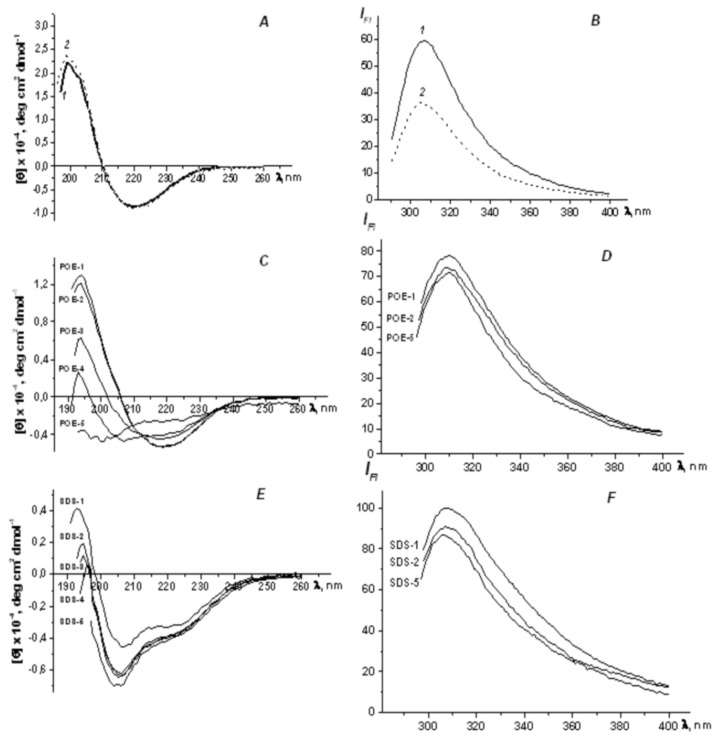
CD spectra in peptide regions (**A**,**C**,**E**) and intrinsic fluorescence spectra atλex 280 nm (**B**,**D**,**F**) of *M. primoryensis* porin (MpOmp) in Tris-HCl buffer(**A**,**B**) and in the presence of POE (**C**,**D**) and SDS (**E**,**F**). The protein samples of MpOmp (1.0 µM and 6.0 µM for peptide and aromatic regions, respectively) were dissolved in 10 mM Tris-HCl buffer, pH 7.5, containing 0.5% POE (1) or 0.25% SDS (2). Samples 2, 3, 4, and 5 were heated at 30, 40, 50, and 80 °C, respectively (**C**,**D**,**E**,**F**).

**Table 1 molecules-25-03131-t001:** Comparison of porin characteristics of some Proteobacteria.

Bacteria	Marinomonas primoryensis KMM 3633^T^ DNA Translation GenBank:QES04118.1	Marinomonas primoryensis KMM 3633^T^ aa Composition (Methods)	Marinomonas primoryensis strain AceL DNA Translation GenBank:WP_112140846.1	Marinomonas sp. IMCC 4694 DNA Translation GenBank:WP_148833815.1	Polaromonas sp. JS666 DNA Translation GenBank:WP_011485260	Magnetospirillum magneticum AMB-1 DNA Translation GenBank:BAE48829.1	Photobacterium damselae subsp. damselae DNA translation GenBank:KJF82573.1	Photobacterium damselae (Vibrio damsela) aa Composition [27]	Escherichia coli DNA Translation GenBank: WP_000768382.1	Escherichia coli aa Sequencing [28]	Basfia succiniciproducens MBEL55E DNA Translation GenBank: AAU37168.1	Average
**Class of Proteobacteria**	Gamma	Gamma	Gamma	Gamma	Beta	Alpha	Gamma	Gamma	Gamma	Gamma	Gamma	
**Superfamily pfam, clan**	Porin_4, cl28788		Porin_4, cl28788	Porin_4, cl28788	Porin_4, cl28788	Porin_4, cl28788	Porin_4, cl28788		OM_channels, cl21487		OM_channels, cl21487	
**Superfamily TCBD**	**Outer membrane pore-forming protein (OMPP) superfamily I**											
**Family TCBD**	The General Bacterial Porin (GBP) Family					The Rhodobacter PorCa Porin (RPP) Family	The General Bacterial Porin (GBP) Family					
**The best match in TCBD, TCID**	1.B.1.1.9 Major outer membrane protein OmpU		1.B.1.1.1 Outer membrane protein F precursor	1.B.1.1.5 Outer membrane porin protein LC PR.	1.B.1.6.1 Outer membrane porin protein 32 pr	1.B.7.1.5 Porin 41 (Por41) OS = Rhodospirillum	1.B.1.1.9 Major outer membrane protein OmpU		1.B.1.1.4 Outer membrane porin protein NMPC		1.B.1.1.14 Major outer membrane protein OS = Pa	
**Query: GenBank:QES04118.1** **BLAST NCBI, Query Cover**	100%		100%	100%	94%	34%	18%		42%		53%	
**Query: GenBank:QES04118.1** **BLAST NCBI, Per. Ident**	100%		83.59%	77.43%	22.42%	24.05%	40.62%		33.90%		33.33%	
**Number of amino acids without signal peptide**	290		299	296	336	412	317		345		345	330
**Molecular weight, kDa**	31.23		31.79	31.73	34.70	42.30	34.94		38.93		38.93	35.57
**Theoretical pI**	3.99		4.17	3.83	9.49	5.73	4.32		4.71		4.71	
**Total number of negatively charged residues (Asp + Glu)**	48		44	50	21	30	54		48		48	
**Total number of positively charged residues (Arg + Lys)**	18		21	15	27	26	31		35		35	
**Aliphatic index:**	63.17		68.49	62.94	60.51	69.88	64.45		61.16		61.16	
**Grand average of hydropathicity (GRAVY)**	−0.491		−0.383	−0.461	−0.362	−0.201	−0.61		−0.624		−0.624	
**Instability index**	−0.01		1.40	10.21	24.19	7.74	15.04		16.18		24.22	

* not determined.

**Table 2 molecules-25-03131-t002:** The amino acid composition of some Proteobacteria porins.

Bacteria	Marinomonas primoryensis KMM 3633^T^ DNA Translation GenBank:QES04118.1	Marinomonas primoryensis KMM 3633^T^ aa Composition (Methods)	Marinomonas primoryensis strain AceL DNA Translation GenBank:WP_112140846.1	Marinomonas sp. IMCC 4694 DNA Translation GenBank:WP_148833815.1	Polaromonas sp. JS666 DNA Translation GenBank:WP_011485260	Magnetospirillum magneticum AMB-1 DNA Translation GenBank:BAE48829.1	Photobacterium damselae Subsp. Damselae DNA Translation GenBank:KJF82573.1	Photobacterium damselae (Vibrio damsela) aa Composition [27]	Escherichia coli DNA Translation GenBank: WP_000768382.1	Escherichia coli aa Sequencing [28]	Basfia succiniciproducens MBEL55E DNA Translation GenBank: AAU37168.1	Average
**Neutral aliphatic, mol %**	**52.1**	**59.8**	**55.5**	**53.7**	**62.6**	**62**	**46.7**	**51.9**	**46**	**50.3**	**46.6**	**53.4**
Gly, mol %	10.3	13	11	10.5	17.6	18	9.8	10.3	11	14.4	11.5	
Ala, mol %	8.3	12.3	10	8.8	10.1	10.2	11.4	10.6	8.4	8.5	7.5	
Val, mol %	9.7	8.3	9.4	7.8	4.2	7.5	4.7	6.9	3.8	6.8	5.6	
Leu, mol %	4.5	5.9	5.7	4.7	5.7	6.6	6	7.9	5.5	6.2	7.1	
Ile, mol %	2.4	3.1	2.3	3.4	4.2	3.2	4.1	4.8	5.2	3.5	4	
Ser, mol %	6.2	8.6	7.4	8.4	11.3	8.7	5	6.2	4.6	4.7	5	
Thr, mol %	10.7	8.6	9.7	10.1	9.5	7.8	5.7	5.2	7.5	6.2	5.9	
**Aromatic, mol %**	**11.4**	**7.3**	**11.3**	**11.5**	**10.2**	**10.0**	**13.6**	**9.7**	**16.3**	**14.2**	**13**	**11.4**
Phe, mol %	4.5	3.4	4.3	4.4	4.5	2.7	6.3	5	5.8	5.6	7.1	
Tyr, mol %	6.9	3.9	7	7.1	4.8	5.1	7.3	4.7	9.3	8.5	5.9	
Trp, mol %	0	- *	0	0	0.9	2.2	0	-	0.6	1.2	1.2	
**Sulphur-containing, mol %**	**1.7**	**0**	**0.7**	**1.7**	**0.9**	**1.9**	**0.9**	**2.8**	**1.7**	**0.9**	**1.2**	**1.3**
Met, mol %	1.7	0	0.7	1.7	0.9	1.9	0.9	2.8	1.7	0.9	1.2	
Cys, mol %	0	0	0	0	0	0	0	0	0	0	0	
**Imino**, mol %	**1.0**	**3.5**	**1.0**	**1.4**	**1.5**	**1.7**	**0.3**	**2.2**	**0.9**	**1.2**	**0.3**	**1.6**
Pro, mol %	1.0	3.5	1.0	1.4	1.5	1.7	0.3	2.2	0.9	1.2	0.3	
**Dicarboxylic, mol %**	**27.3**	**20.3**	**24.4**	**26.9**	**15.9**	**17**	**28.8**	**21.8**	**24.4**	**24**	**21.2**	**22.9**
Asx, mol %	18.3	12	15.4	16.2	9.9	11.2	18.7	14.4	17.1	16.7	12.5	
Glx, mol %	9	8.3	9	10.7	6	5.8	10.1	7.4	7.3	7.3	8.7	
**Basic, mol %**	**6.5**	**7.4**	**7**	**5.1**	**7.7**	**7.5**	**9.8**	**11.5**	**10.7**	**8.8**	**16.4**	**8.9**
His, mol %	0.3	1	0	0	1.2	1.2	0	1.6	0.6	0.3	0.9	
Arg, mol %	0.7	2.1	1	1	6.5	2.4	2.5	4.2	4.6	3.2	5.3	
Lys, mol %	5.5	4.3	6	4.1	1.5	3.9	7.3	5.7	5.5	5.3	10.2	
**Polarity index (Hatch)**	**1.95**	**1.81**	**2.07**	**2.16**	**2.13**	**1.69**	**2.21**	**1.46**	**2.03**	**1.79**	**1.88**	**1.93**
**Polarity (Nitzan)**	**50.7**	**44.25**	**48.95**	**50.05**	**46.1**	**43.7**	**52.5**	**45.7**	**51**	**49.55**	**51.3**	**48.5**

**Table 3 molecules-25-03131-t003:** Content of secondary structural elements in porin from *M. primoryensis* KMM 3633^T^ (%).

№	Sample	α-Helix			β-Structure			β-Turn	Random Coil
		I	II	III	I	II	III		
**1**	MpOmp	**2.8**	1.5	**4.3**	32.5	17.4	**49.9**	**20.3**	**25.5**
**2**	MpOmp, 0.5% POE	0.8	4.1	**4.9**	34.7	14.4	**49.1**	**20.9**	**25.1**
**3**	MpOmp, 0.25% SDS	7.6	10.4	**18.0**	18.1	10.2	**28.3**	**22.7**	**31.0**

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
