# Peer review of "Porin from Marine Bacterium Marinomonas primoryensis KMM 3633T: Isolation, Physico-Chemical Properties, and Functional Activity"

_molecules, 2020, doi:10.3390/molecules25143131_

Round 1

Reviewer 1 Report

The manuscript presents interesting data on the outer membrane porin protein of the Gram-negative marine bacterium Marinomonas primoryensis. The work can bring an important contribution to the characterization of Marinomonas primoryensis proteins, beyond being relevant to what respects extraction and purification of porin proteins.

Nonetheless, the manuscript presents some major (scientific) and minor (formatting) structural issues that need to be addressed, namely (Please note: line numbering is absent from page 6 onwards):

A. Scientific issues

A.1. Results - Section 2.1: The authors describe a “temperature denaturation” effect based on the SDS-PAGE data solely. Nonetheless, three different aspects need to be considered regarding this data:

I) authors need to differentiate the process of dissociation from the process of denaturation. Oligomeric proteins dissociate at lower concentrations. Was the protein diluted from lines 1 and 6 (Fig.1C) to lines 2-5 for sarcosyl and 7-10 for POE to carry out the experiments at different temperatures? What were the concentrations used? Regarding oligomeric proteins, dilution is enough to dissociate trimers to monomers, even at room temperature;

II) authors need to use a more accurate and direct method to confirm/corroborate the temperature effect reported by SDS-PAGE (e.g. monitor the effect of temperature by CD or fluorescence spectroscopy);

III) authors also need to monitor the process without SDS. SDS is by itself a detergent that can also be influencing the dissociation process and oligomerization state of the protein. Thus, methods like gel filtration chromatography can be applied, with the column properly equilibrated with Sarcosyl and POE solely, in order to avoid the presence of the SDS-extra detergent and identify the true oligomerization state of the protein (distribution of protein species among trimers or monomers).

A.2. Results - Section 2.3 should appear earlier in the manuscript, possibly as section 2.2 in order to firstly present the amino acid sequencing data (confirming that the authors are dealing with the correct protein) before moving to structural and functional aspects. Which BLAST tool was used for the analysis? No reference is introduced for BLAST in the section. The acronym TBCD needs to be defined in the text (line 217). Lines 230 and 231 need to be rephrased to “(…) calculated as reported previously [29]”.

A.3. Results - Section 2.4.1 Once again authors mention a “denaturation” process, when in fact the far UV CD data (Fig.3B) is reporting a conformational change/transition from a typical beta-sheet spectrum (1) to an alpha-helix spectrum (2). So, in fact what we seem to be assisting (and it is interesting to explore and report) is to a structural transition from secondary beta-sheet (in the presence of POE) to alpha-helix (in the presence of SDS) (also confirmed by the percentages given by the CDPro algorithm in Table 2). Denatured/random coil protein structures show very different far UV CD profiles from the one in Fig.3. So, in 0.25% SDS, the protein may dissociate and show more alpha-helical content that stabilizes the monomeric species. Indeed, this can also justify the observations seen by SDS-PAGE (where SDS is in fact at a higher percentage than in the SDS-PAGE gel and electrophoretic buffer). The amino acid composition (positive, negative, polar vs non-polar amino acids) and the hydrophobic and electrostatic interactions protein-protein and protein-solvent are certainly playing a role and governing the beta-sheet to alpha-helix transition, when using different solvents (either POE, SDS or sarcosyl). These aspects (that authors mention lightly at the end of the Results section), should be studied are discussed more deeply. Is the beta-sheet to alpha-helix transition reversible? How is protein structure influenced by detergent charge and detergent concentration? How does it correlate with charge and amino acid distribution? What is the oligomerization state of the protein under these different conditions? How is the pore-forming capacity affected afterwards?

A4. Sentences in lines 144 and 159 need a supporting reference.

A5. What was the excitation wavelength used in the fluorescence experiments shown in Fig.3C? Authors report on the Materials and Methods section that three wavelengths have been tested (Page.17) but given the absence of tryptophan moieties in the protein, the intrinsic fluorescence shown by the protein is certainly very different when using different excitation wavelengths.

B. Formatting issues

B.1 English language and style

The manuscript needs to undergo revision by an English native speaker. Please correct the errors and grammar faults that appear throughout the text (singular versus plurals, verbs, adverbs, prepositions, pronouns, punctuation and phrasing).

B.2 Taxonomy – Scientific names in italic

Please verified the typing style applied throughout the text to scientific designations (bacterial species and families). Some names are not in italic (e.g.: line 78 – Enterobacteriaceae; lines 116 and 174 - M. primoryensis; lines 209 and 210 – Gammaproteobacteria), line 215-Proteobacteria.

B.3 Font, line spacing and Figures and tables

Please apply the same font type and size as well as the proper line spacing throughout the text. Several sections are not properly formatted. Figure captions (Fig.1, Fig2 and Fig 3) are also not conveniently formatted and appear mixed with the main text.

Figure 1: Consider labelling the bottom region of Fig.1 with the gel’s sections to differentiate the two extractions with Sarcosyl and Octyl-POE. Panels A to C are nor properly differentiated in the legend of Fig.1.

Figure 3: Please use the same numbering in Fig. 3 and Table 2 in order to facilitate reading and correlation (1-POE and 2-SDS or the other way around). Please use darker colours for Fig.3A to stand out the near UV CD spectra. Please reduce the size of Fig.3 so it fits in a row, with 3 panels longitudinally (panels can be smaller, since spectra present enough resolution).

Table 1: Please remove the typing errors and reduce the size of table 1 in order to present the information more concisely in 1 or 2 pages, showing only the most relevant data. Some information can be presented in one or more tables separately, or as supplementary material (for example, amino acid distributions can be organised in a different table). Table legends can also be used to describe acronyms in full.

Author Response

Response to Reviewer 1 Comments

Point 1: A.1. Results - Section 2.1: The authors describe a “temperature denaturation” effect based on the SDS-PAGE data solely. Nonetheless, three different aspects need to be considered regarding this data:A.1. Results - Section 2.1: I) authors need to differentiate the process of dissociation from the process of denaturation. Oligomeric proteins dissociate at lower concentrations. Was the protein diluted from lines 1 and 6 (Fig.1C) to lines 2-5 for sarcosyl and 7-10 for POE to carry out the experiments at different temperatures? What
were the concentrations used? Regarding oligomeric proteins, dilution is enough to dissociate trimers to monomers, even at room temperature;

Response 1: Of course, the sensitivity of individual proteins to denaturing agents is different. However, porin proteins belong to membrane proteins, which determines the need to use detergents to solubilize them. The dissociation of trimers of these proteins occurs only under the influence of temperature. The range of values of this so-called critical temperature of the irreversible conformational transition accompanying the dissociation into monomers is quite
wide. For known porins of gram-negative bacteria, it is 50-80 ° C. In the case of the protein isolated by us (MpOmp), this value is abnormally low. Therefore, taking into account the comments of the reviewer 1, we recorded the spectra characterizing the spatial structure of the protein (CD in the peptide region and the total fluorescence spectrum) for two protein concentrations in a buffer without detergent. Figure and discussion of the results are included in the revised manuscript. As follows from the data in the figure, no conformational changes are observed either at the level of the secondary or at the level of the tertiary structure of the protein upon dilution. In this regard, when determining the critical temperature of the thermal transition in the dilution of protein samples was not necessary.

Point 2: II) authors need to use a more accurate and direct method to confirm/corroborate the temperature effect reported by SDS-PAGE (e.g. monitor the effect of temperature by CD or fluorescence spectroscopy); Point 3: III) authors also need to monitor the process without SDS. SDS is by itself a detergent that can also be influencing the dissociation process and oligomerization state of the protein. Thus, methods like gel filtration
chromatography can be applied, with the column properly equilibrated with Sarcosyl and POE solely, in order to avoid the presence of the SDS-extra detergent and identify the true oligomerization state of the protein (distribution of protein species among trimers or monomers).

Response 2 and 3: In accordance with the recommendations of reviewer 1, the process of denaturation of MpOmp under the influence of temperature was studied using optical methods. In connection with this, Figure 3 is completely redone and the corresponding additions are included in the revised manuscript (2.4.2. Temperature stability of OM porin from M. primoryensis 3.3. Physico-chemical properties of porin from M. primoryensis).

Point 4: A.2. Results - Section 2.3 should appear earlier in the manuscript, possibly as section 2.2 in order to firstly present the amino acid sequencing data (confirming that the authors are dealing with the correct protein) before moving to structural and functional aspects. Which BLAST tool was used for the analysis? No reference is introduced for BLAST in the section. The acronym TBCD needs to be defined in the text (line 217). Lines 230 and 231 need to be rephrased to “(…) calculated as reported previously [29]

Response 4: We used 2 data bases of the protein sequences for the BLAST analysis, as demonstrated in the Table 1 and in the Methods. The first data base was GebBank NCBI, which is the biggest non-specialized database associated with databases EMBL and KEGG. The Query was the protein sequence of the porin of Marinomonas primoryensis KMM 3633T (GenBank accession number QES04118.1). This Query is indicated in the Table 1. Query (QES04118.1) covering and identity were used for similarities characteristics. The second data base was the specialized database TCBD - Transporter Classification Data Base. The acronym TBCD was defined in the Methods. Corrections included in the revised manuscript

Point 5: A.3. Results - Section 2.4.1 Once again authors mention a “denaturation” process, when in fact the far UV CD data (Fig.3B) is reporting a conformational change/transition from a typical beta-sheet spectrum (1) to an alpha-helix spectrum (2). So, in fact what we seem to be assisting (and it is interesting to explore and report) is to a structural transition from secondary beta-sheet (in the presence of POE) to alpha-helix (in the presence of SDS) (also confirmed by the percentages given by the CDPro algorithm in Table 2). Denatured/random coil protein structures show very different far UV CD profiles from the one in Fig.3. So, in 0.25% SDS, the protein may dissociate and show more alpha-helical content that stabilizes the monomeric species. Indeed, this can also justify the observations seen by SDS-PAGE (where SDS is in fact at a higher percentage than in the SDS-PAGE gel and electrophoretic buffer). The amino acid composition (positive, negative, polar vs nonpolar amino acids) and the hydrophobic and electrostatic interactions protein-protein and protein-solvent are certainly playing a role and governing the beta-sheet to alpha-helix transition, when using different solvents (either POE, SDS or sarcosyl). These aspects (that authors mention lightly at the end of the Results section), should be studied are discussed more deeply. Is the beta-sheet to alpha-helix transition reversible? How is protein structure influenced by detergent charge and detergent concentration? How does it correlate with charge and amino acid distribution? What is the oligomerization state of the protein under these different conditions? How is the pore-forming capacity affected afterwards?

Response 5: The issues raised by reviewer 1 in this section of review are certainly important and interesting. Regarding porin proteins, there is conflicting information in the literature on dissociation-denaturation. Some authors have shown the loss of the trimeric structure of porin only after the deployment of monomeric subunits. Other researchers initially observed the dissociation of porin oligomers into folded monomers. Therefore, for a definite answer to these
questions, a special study is required, which we plan in the future. The experience that we have in the study of porin dissociation-denaturation allows us to approach these aspects deeply and seriously [Olga D. Novikova Dmitry K. Chistyulin,a Valentina A. Khomenko,a Evgeny V. Sidorin,a Natalya Yu. Kim,a Nina M. Sanina, b Olga Yu. Portnyagina,a Tamara F. Solov'eva,a Vladimir N. Uversky and Valery L. Shnyrov Peculiarities of thermal denaturation of OmpF porin from Yersinia ruckeri Mol Biosyst 2017 Aug 22;13(9):1854-1862. doi:10.1039/c7mb00239d.]

Point 6: A4. Sentences in lines 144 and 159 need a supporting reference.
Response 6: Corrections included in the revised manuscript

Point 7: A5. What was the excitation wavelength used in the fluorescence experiments shown in Fig.3C? Authors report on the Materials and Methods section that three wavelengths have been tested (Page.17) but given the absence of tryptophan moieties in the protein, the intrinsic fluorescence shown by the protein is certainly very different when using different excitation wavelengths.
Response 7: Corrections included in the revised manuscript

Point 8: B. Formatting issues. B.1 English language and style
Response 8: In the case of acceptance of our article for publication, we would use the services of specialists of MDPI English editing service.

Point 9: B.2 Taxonomy – Scientific names in italic
Response 9: Corrections included in the revised manuscript

Point 10: B.3 Font, line spacing and Figures and tables
Response 10: Corrections included in the revised manuscript

Point 11: Figure1 :
Response 11: Figure 1C fixed

Point 12: Figure3 :
Response 12: Figure 3 is completely redone and corresponding explanations are included in the text of the revised manuscript

Point 13: Table1:
Response 13: It seems to us that the data given in the table are of interest in full. Considering the reviewer's recommendations, we split them into two tables.

Reviewer 2 Report

The authors in this manuscript report exciting data ion channels from bacteria. The topic is attractive as well as the results, the article needs to be revised in some parts to increase the quality.

The topic is attractive as well as the results, although the article needs to be revised because it lacks clarity. Furthermore, in addition to the hypothesis reported concerning the content of secondary structures in the presence of ionic and non-ionic detergents (denaturing effect), the hypothesis that SDS micelles favor the formation of alpha helixes and non-ionic beta-sheets as widely observed for intrinsically disordered proteins (see doi:10.1016/j.bbamem.2018.02.022).

Chapter 3.2 [Error! Reference source not found.] probably a reference is missing.

Regarding the formation of quaternary structures, a theory has recently been reported that explains the formation of multimeric proteins. It has been reported that the formation of multimeric proteins is due to the balance between the intra-molecular interaction energy and the adhesion energy between subunits. If intra-molecular energy is greater than the adhesion energy (monomer-monomer), oligomers are formed, while if the intra-molecular is higher than adhesion energy, protein remains as monomer. This concept should be discussed (see doi:10.1007/s0024 9-020-01424 -1).

Author Response

Response to Reviewer 2 Comments
Point 1: Furthermore, in addition to the hypothesis reported concerning the content of secondary structures in the presence of ionic and non-ionic detergents (denaturing effect), the hypothesis that SDS micelles favor the formation of alpha helixes and non-ionic beta-sheets as widely observed for intrinsically disordered proteins (see doi:10.1016/j.bbamem.2018.02.022).
Response 1: A reference that recommended by reviewer 2 was used when discussing the results (Discussion 3.3. Physico-chemical properties of porin from M. primoryensis) in the revised manuscript. We propose a detailed study of changes in the spatial structure of MpOmp under the influence of temperature in solutions of ionic and nonionic detergents.

Point 2: Chapter 3.2 [Error! Reference source not found.] probably a reference is missing.
Response 2: Corrections included in the revised manuscript

Point 3: Regarding the formation of quaternary structures, a theory has recently been reported that explains the formation of multimeric proteins. It has been reported that the formation of multimeric proteins is due to the balance between the intra-molecular interaction energy and the adhesion energy between subunits. If intra-molecular energy is greater than the adhesion energy
(monomer-monomer), oligomers are formed, while if the intra-molecular is higher than adhesion energy, protein remains as monomer. This concept should be discussed (see doi:10.1007/s00249-020-01424 -1).
Response 3: The concept proposed by the reviewer 2 regarding the formation of multimeric proteins is mentioned in the discussion of the instability of the trimeric form of the studied porin. Corresponding addition is included in the text of the revised manuscript . It should be noted that, despite the fact that the MpOmp trimer is unstable, as evidenced by the low critical temperature of irreversible thermal transition, in the protein solutions obtained during the
isolation and purification of porin, the formation of amyloid-like structures is not observed.

Round 2

Reviewer 1 Report

The manuscript has been considerable reorganized and improved. Nonetheless, the following aspects still need adjustment:

  1. Please use capital letters in “Gram” as in Gram-negative throughout the text.
  2. Please use italic in for Proteobacteria and Marinomonas throughout the text.
  3. Please replace the comma by a point in 0.8 m (0,8 m in Amursky Bay near Vladivostok – Page 4)
  4. Please replace the verb “to give” by “to describe” in ((…) the growth medium described previously [23].) – Page 4
  5. Please remove the “Error Messages” in Tables 1 and 2 and adjust the formatting.
  6. Please remove the “nm” in duplicate in (209 nm nm) - Page 14
  7. Please mention the range of temperatures tested in the presence of SDS and POE in the legend of Figure 3 (panels A to F).
  8. It might be quite farfetched/risky to assume that no changes have occurred at the tertiary level, especially when secondary structure changes are evident by far UV CD. Even if near-UV CD and fluorescence spectroscopy are more sensitive to changes in the tertiary structure of proteins, due to changes in the chemical environment of the aromatic moieties (mainly of Tyr in the case of Porin), they may just not be noticeable by fluorescence, as the authors mention (only some decrease in fluorescence intensity is observed, probably due to quenching effects resulting for the increase in temperature). Tyr residues may not be suffering and/or reporting changes in their chemical environment, especially when using 280 nm as excitation wavelength where the Anti-Stokes/Raman Scattering can also be contributing to the signal at 310 nm. So, it would be better to rewrite the phrase to something like “(…) did not lead to any major noticeable changes detected by fluorescence spectroscopy.” – Page 17.
  9. Reference formatting needs to be properly adjusted.

Author Response

Response to Reviewer 1 comments

    Point 1: Please use capital letters in “Gram” as in Gram-negative throughout the text.

    Response 1: Corrections are included in the revised manuscript.

    Point 2: Please use italic in for Proteobacteria and Marinomonas throughout the text.

    Response 2: Taxonomic names were written using italic.

   Point 3:  Please replace the comma by a point in 0.8 m (0,8 m in Amursky Bay near Vladivostok – Page 4)

Response 3: “0,8 m” was replaced by “0.8 m”.

   Point 4:  Please replace the verb “to give” by “to describe” in ((…) the growth medium described previously [23].) – Page 4

Response 4: “to give” was replaced by “to describe”.

     Point 5:    Please remove the “Error Messages” in Tables 1 and 2 and adjust the formatting.

Response 5: “Error Messages” was removed.

   Point 6:     Please remove the “nm” in duplicate in (209 nm nm) - Page 14

    Response 6: “nm” was removed.

Point 7:    Please mention the range of temperatures tested in the presence of SDS and POE in the legend of Figure 3 (panels A to F).

Response 7: The range of temperatures in the legend of Figure 3 (panels C- F) was shown.

  Point 8:      It might be quite farfetched/risky to assume that no changes have occurred at the tertiary level, especially when secondary structure changes are evident by far UV CD. Even if near-UV CD and fluorescence spectroscopy are more sensitive to changes in the tertiary structure of proteins, due to changes in the chemical environment of the aromatic moieties (mainly of Tyr in the case of Porin), they may just not be noticeable by fluorescence, as the authors mention (only some decrease in fluorescence intensity is observed, probably due to quenching effects resulting for the increase in temperature). Tyr residues may not be suffering and/or reporting changes in their chemical environment, especially when using 280 nm as excitation wavelength where the Anti-Stokes/Raman Scattering can also be contributing to the signal at 310 nm. So, it would be better to rewrite the phrase to something like “(…) did not lead to any major noticeable changes detected by fluorescence spectroscopy.” – Page 17.

Response 8: Appropriate replacement in the text (page 17) was done according comment of Reviewer 1.

Point 9:  Reference formatting needs to be properly adjusted.

Response 9:  Reference formatting was done.